# Dysglycemia and the airway microbiome in cystic fibrosis

Laura N. Brenner[1,2]*, Ching-Ying Huang[1,2], Minsik Kim[1,2], Lindsay Bringhurst[1,2,3], Christopher J. Richards[1,2,3], Leonard Sicilian[1,2,3], Isabel Neuringer[1,2,3], Melissa S. Putman[2,3,4], Peggy S. Lai[1,2]

1 Division of Pulmonary and Critical Care Medicine, Massachusetts General Hospital, Boston, Massachusetts, United States of America, 2 Department of Medicine, Massachusetts General Hospital, Boston, Massachusetts, United States of America, 3 Cystic Fibrosis Center, Massachusetts General Hospital, Boston, Massachusetts, United States of America, 4 Diabetes Research Center, Massachusetts General Hospital, Boston, Massachusetts, United States of America

* Lnbrenner@mgh.harvard.edu

## Abstract

### Background

Cystic fibrosis-related diabetes (CFRD) is one of the most common non-pulmonary complications in people living with cystic fibrosis (pwCF), seen in up to 50% of adults. Even when correcting for severity of *CFTR* mutations, those with CFRD have more pulmonary exacerbations, lower lung function, and increased mortality than those with normal glucose tolerance (NGT).

### Methods

Expectorated sputum samples were collected from 63 pwCF during routine outpatient visits (29 with CFRD, 12 with IGT and 22 with NGT). Oral glucose tolerance test results, A1c levels, and pulmonary function tests closest to the time of sputum collection were obtained from the medical record. Samples underwent metagenomics sequencing and raw reads were processed through the bioBakery workflow for taxonomic profiling at the species level as well as predicted functional profiling and antibiotic resistance profiling. Viral profiling was performed with Marker-MAGu. Differences in alpha diversity, beta diversity, and differential abundance were assessed. Microbiome and phage signatures of CFRD were generated using sparse partial least squares models which were subsequently used as a primary predictor of lung function using multivariate linear regression.

### Results

In linear models, CFRD status compared to NGT was associated with a lower alpha diversity (reciprocal Simpson −1.98 [−3.80,-0.16], $p = 0.033$) and differences in microbial community composition (Bray Curtis dissimilarity PERMANOVA $R^2$ 0.17,

**Data availability statement:** Raw sequencing data generated and analyzed in this study have been deposited in the NCBI Sequence Read Archive (SRA) under accession numbers PRJNA861321 and PRJNA1078869. The datasets are available at: • https://www.ncbi.nlm.nih.gov/sra/?term=PRJNA861321 • https://www.ncbi.nlm.nih.gov/sra/?term=PRJNA1078869.

**Funding:** L.N.B. is supported by the NIDDK K23 DK125839 and the Cystic Fibrosis Foundation (CFF) K Boost Award. There was no additional external funding received for this study.

**Competing interests:** The authors have declared that no competing interests exist.

$p = 0.011$). *Pseudomonas aeruginosa* and Streptococcus gordonii had higher relative abundance in CRFD vs NGT participants (2.43 [0.027, 4.82], unadjusted $p = 0.056$ and 1.11 [0.58, 1.64] unadjusted p=<.001 respectively). There were global differences between CFRD vs NGT in both functional pathways and antibiotic resistance genes. In multivariate models adjusting for age, sex, antibiotic use, and modulator therapies, virome but not microbiome signatures of CFRD were associated with lower FEV1 percent predicted (−6.4 [95% CI −10.2, −2.6]%, $p = 0.001$ for each 10% increase in virome score).

## Conclusion

Differences in the airway microbiome in those with dysglycemia in CF are associated with poorer lung function.

## Introduction

Cystic fibrosis (CF) is characterized by abnormal mucus production leading to impaired mucociliary clearance in the respiratory tract, resulting in chronic inflammation and colonization by microorganisms. This process forms a distinct complex ecosystem within the airways known as the respiratory microbiome. The sputum microbiome in CF is distinct from that of healthy individuals, exhibiting dysbiosis characterized by reduced microbial diversity and altered community composition [1].

Cystic fibrosis-related diabetes (CFRD) is one of the most common non-pulmonary complications of CF, seen in up to 50% of adults [2]. The etiology of CFRD is multifactorial, involving both genetic and environmental factors. Even accounting for cystic fibrosis transmembrane conductance regulator (*CFTR*) mutation severity, those with CFRD have more pulmonary exacerbations, lower lung function, and higher mortality than those with normal glucose tolerance (NGT) [3]. Early diagnosis of CFRD and treatment with insulin lead to improved clinical outcomes including increased pulmonary function, higher BMI, and decreased mortality [3]. Adults with CFRD have increased culture growth of *Staphylococcus aureus* and *Pseudomonas aeruginosa* compared to those without CFRD. Co-infection with these two pathogens is associated with decreased lung function and increased pulmonary exacerbations in people with CFRD compared to those with NGT [4].

Dysglycemia in people both with and without CF has been associated with changes in respiratory and stool bacteria. An increase in serum glucose has been shown to increase both upper and lower respiratory glucose levels in people with and without CF [5–7]. As bacteria are known to use glucose as a major source of energy, it is not surprising that hyperglycemia is related to high abundance of respiratory pathogens in people without CF [8]. Glycemia has been shown to be associated with the composition and microbial diversity of the microbiome in various body sites including the respiratory and GI tract in those without CF [9]; however, little is known how diabetes impacts the airway microbiome in people with CF (pwCF). A recent study of 30 individuals with CF showed a change in microbiome composition

in people with CFRD. 16S rRNA sequencing was used to show a genus-level increase in *Pseudomonas*, *Azihorizophilus*, *Porphyromonas* and *Acintobactillis*, but metagenomic sequencing allowing for finer taxonomic resolution and characterization of phage communities has not previously been done.

The goal of this study was to characterize differences in the airway microbiome and phageome in adults with CF with and without CFRD and further to determine if microbial signatures of dysglycemia are associated with lung function.

## Methods

### Study design

We recruited adults with CF receiving care at the Massachusetts General Hospital Adult Cystic Fibrosis Center. Spontaneously expectorated sputum was collected during routine outpatient visits between 2019 and 2021 in pwCF who were not having an exacerbation (only 9% of samples were collected after March 2020). The samples were aliquoted and frozen at -80C until nucleic acid extraction. The study received ethical approval by the Institutional Review Board of Mass General Brigham (Protocol # 2019P002868) and written informed consent was obtained from all participants. CFRD status was defined based on a standardized oral glucose tolerance test (stratified into normal glucose tolerance [NGT], impaired glucose tolerance [IGT], or CFRD). The most recent pulmonary function test from the sample collection date was obtained from the medical record. Pulmonary function percent predicted (FEV1 and FVC) was calculated using the Global Lung Initiative race-neutral equations [10]. Antibiotic use was defined as any inhaled or systemic antibiotic use at the time of sputum collection.

### Nucleic acid extraction

Samples, reagent-only negative controls, and mock community-positive controls (Zymo Research D6300) were extracted using a protocol optimized for respiratory samples with a magnetic bead-based protocol using the Maxwell HT 96 gDNA Blood Isolation System (Promega) on a KingFisher Flex instrument as previously described (see S1 File) [11].

### qPCR microbial load analysis

Quantitative PCR (qPCR) targeting the 16S rRNA gene was performed on sputum samples using a TaqMan probe with FAM dye for detection (Thermo Fisher 16S Pan-bacterial Control (Assay ID Ba04230899_s1). Bacterial standards generated from *E. coli* strain JM109 (Zymo Catalog # E2006) were used to create a standard curve for DNA quantification. All qPCR reactions were performed in triplicate using the QuantStudio 7 Pro (Life Technologies). The reaction conditions were initial 2 min at 25°C, 15 min at 50°C, and 2 min at 95°C, then 40 cycles of 95°C for 3 seconds (denaturation) and 55°C for 30 seconds (annealing). QuantStudio Design & Analysis 2.8 (Thermo Fisher Scientific) was used for qPCR analysis.

### Bioinformatics processing

Raw data files in binary base call (BCL) format were converted into FastQ files and demultiplexed based on the dual-index barcodes using the Illumina "bcl2fastq" software, then subsequently processed using bioBakery3 [12]. KneadData was used to process demultiplexed files. Trimmomatic [13] was used to remove human sequences, low-complexity and repetitive sequences, and adapter and low-quality bases with, and contaminant checks were done with bowtie2 [14], Processed FastQ reads were first mapped against the MetaPhlAn4 [15] marker gene database (mpa v30 CHOCOPhlAn 201901) to generate taxonomic profiles per sample at the species level.

Functional profiling of the microbial community was performed using HUMAnN3 [16] and binned to the BioCyc [17] pathway database, resulting in gene family abundance tables assembled into higher order Kegg pathways [18]. Antimicrobial resistance gene marker gene sequences were obtained from the Comprehensive Antibiotic Resistance Database (CARD) version 3.0.7 [19] and further annotated with MEGARes [19,20]. Antimicrobial resistance profiles were then

generated for each sample with ShortBRED [21] using these databases as the references. Microbial profiles and sample metadata were combined into phyloseq objects [22] for analysis. MarkerMAGu [23] was utilized to identify phage sequences by mapping contigs against a curated database of viral marker genes. Phage abundance was quantified by normalizing the read counts mapped to these markers, providing an estimation of phage presence and relative abundance across samples. Taxonomic assignment of the identified phages was performed using MarkerMAGu's integrated classification system, which leverages both viral marker genes and nucleotide sequences.

### Statistical analyses

Participants with diagnosed CFRD (n = 29) were compared to participants with NGT (n = 22). Those with IGT were included in diversity analyses, but excluded in abundance analyses as IGT is considered a distinct entity to NGT and CFRD and not an intermediary [24]. Age, biological sex, antibiotic use, and CFTR modulator use were included in all models as covariates.

To test the hypothesis that alpha diversity (species richness, inverse Simpson) differed by CFRD status, we performed linear regression adjusting for covariates. To test the hypothesis that CFRD status was associated with overall microbial community structure, we performed permutational multivariate analysis of variance (PERMANOVA) using Bray-Curtis and Horn-Morisita dissimilarity metrics to assess differences in community composition while adjusting for covariates. Differential abundance testing of microbes and pathways associated with CFRD status was performed with linear models using centered log-ratio transformed relative abundance for each microbe, adjusting for covariates. Multiple testing was accounted for with a false discovery rate <0.10 considered statistically significant.

Microbiome and virome signatures were created using sparse Partial Least Squares (sPLS) regression to classify sputum samples based on CFRD status using the centered log-ratio (CLR) transformed microbial relative abundance. The sPLS model was trained using mixOmicsCaret [25,26] integration, optimizing the number of components and feature selection parameters through grid search and cross-validation. CV fold number was chosen using lowest RMSE value and highest $R^2$ value. The best-performing repetition of the model was identified, and the cross-validated AUC was calculated to provide a robust estimate of the model's ability to distinguish between CFRD-positive and negative samples. The relationship between the microbiome and virome signatures on lung function was assessed jointly using multivariate linear regression, adjusting for age, biological sex, CFTR modulator use, antibiotic use, and CFRD status.

All analyses were performed in R version 4.4.1.

## Results

### Participant Characteristics

A total of 63 participants were included in this study, with 29, 12, and 22 classified as having CFRD, IGT, and NGT. Mean age was 41 +/- 13.1 years old, 43.5% were males. Seventy-five percent of samples were collected while the participant was taking a CFTR modulator (Table 1). PFTs were performed a median 0.0 [IQR 0.0, 0.0] days before sample. A1c was taken a median 79.5 [IQR 0, 271] days before sample collection.

### Read depth

Metagenomic sequencing yielded a median of 84.5 million [Interquartile range 51.1–123.1 million] reads per sample after removal of low quality and host (human) reads.

### Taxonomic analysis

In our linear model assessing the relationship between glycemic status and alpha diversity, CFRD status was associated with lower alpha diversity, as reflected by a decrease in the Inverse Simpson index (−1.98 [−3.80, −0.16], p = 0.033) and a

**Table 1. Participants Characteristics.**

| | CFRD | IGT | NGT | p |
|---|---|---|---|---|
| n | 29 | 12 | 22 | |
| Age (years) | 40.43 (11.41) | 33.31 (10.47) | 43.54 (15.16) | 0.088 |
| Male sex | 10 (34.5) | 8 (66.7) | 9 (40.9) | 0.162 |
| BMI (kg/m$^2$) | 22.01 (2.82) | 23.74 (2.10) | 24.41 (3.89) | 0.025 |
| HgbA1c (%) | 6.82 (1.38) | 5.55 (0.50) | 5.52 (0.37) | <0.001 |
| Pancreatic insufficiency | 29 (100.0) | 12 (100.0) | 11 (50.0) | <0.001 |
| CFTR modulator use | 24 (82) | 10 (83.3) | 11 (50.0) | 0.022 |
| FEV1% predicted | 50.54 (25.35) | 77.54 (17.93) | 63.21 (24.52) | 0.005 |
| Antibiotic use | 27 (93.1) | 9 (75) | 16 (72.2) | 0.123 |

Data are presented as mean (SD) or n (%) unless otherwise specified.

CFRD, cystic fibrosis related diabetes; IGT, impaired glucose tolerance; NGT, normal glucose tolerance; CFTR, cystic fibrosis transmembrane conductance regulator; FEV1, forced expiratory volume in one sec, as calculated by the GLI 2022 race neutral equations.

nonsignificant reduction in species richness (−3.14 [−7.61, 1.34], p = 0.17). In the multivariate model, the effect size of the association between CFRD status and the Inverse Simpson index remained similar, with a beta estimate of −2.00 [−4.02, 0.02] (p = 0.052). IGT status was not significantly associated with alpha diversity. To test the hypothesis that glycemia affected overall microbial community structure, we assessed differences in beta diversity. CFRD status was associated with microbial community structure as measured by the Bray Curtis (PERMANOVA p = 0.011, R$^2$ = 17%) (Fig 1) and Horn-Morisita (PERMANOVA p = 0.0067, R$^2$ = 19%) dissimilarity in the multivariate model.

## Differential Abundance comparisons

To investigate what taxa were driving the diversity differences, we focused on differential abundances between CFRD and NGT. After CLR transformation of relative abundance CFRD was associated with a 2.43 [0.027, 4.82] log fold increase in *Pseudomonas aeruginosa* (unadjusted p = 0.056, q = 0.449) and a 1.11 [0.58, 1.64] increase in *Streptococcus Gordonii* (unadjusted p < 0.001, q = 0.01) see Fig 2 and S1 Table.

## Bacterial load using 16S rRNA qPCR

Mean 16S rRNA gene concentration was 4.32 +/- 3.85 ng/μL in samples from pwCFRD and 4.45 +/- 4.38 in samples from pwNGT. In a multivariate model including CFRD status, age, antibiotic use, and sex, CFRD status did not show statistically significant associations with 16S rRNA concentration (regression coefficient = 0.265 [−1.77, 2.30] p = 0.80).

## Functional pathway analysis

There were differences in functional pathway measured by Bray-curtis (PERMANOVA R$^2$ = 0.20, p = 0.002) when comparing different statuses of glycemia in multivariate models. Differential abundance testing demonstrated statistically significant differences in multiple pathways when comparing samples from pwCFRD and those with NGT (Fig 3, S2 Table). Pathways associated with tyrosine metabolism were associated with CFRD status with 4.54 [2.65, 6.43] log fold increased in the sputum microbiome in people with CFRD vs those with NGT (FDR q-value = 0.002).

## Antimicrobial resistance analysis

There were differences in antibiotic resistance genes when comparing CFRD status (NGT, IGT, and CFRD) (Bray-curtis PERMANOVA R$^2$ = 0.15, p = 0.0049). The presence of CFRD was closely linked with differences in the

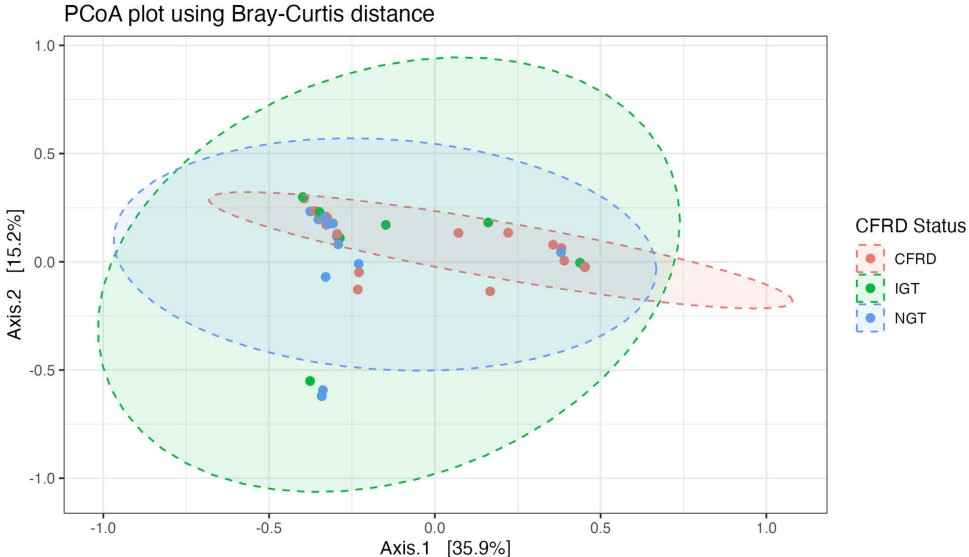

**Fig 1. Principal Coordinates Analysis (PCoA) Plot of Beta Diversity Among CFRD, IGT, and NGT Groups Based on Bray-Curtis Dissimilarity.**
PCoA plot illustrating the beta diversity of the microbial communities among individuals with cystic fibrosis-related diabetes (CFRD), impaired glucose tolerance (IGT), and normal glucose tolerance (NGT). Each point represents a sample, colored by CFRD status. The distances between points reflect the compositional dissimilarities in microbial communities, with closer points indicating more similar microbiomes. Ellipses represent the 95% confidence intervals for each group.

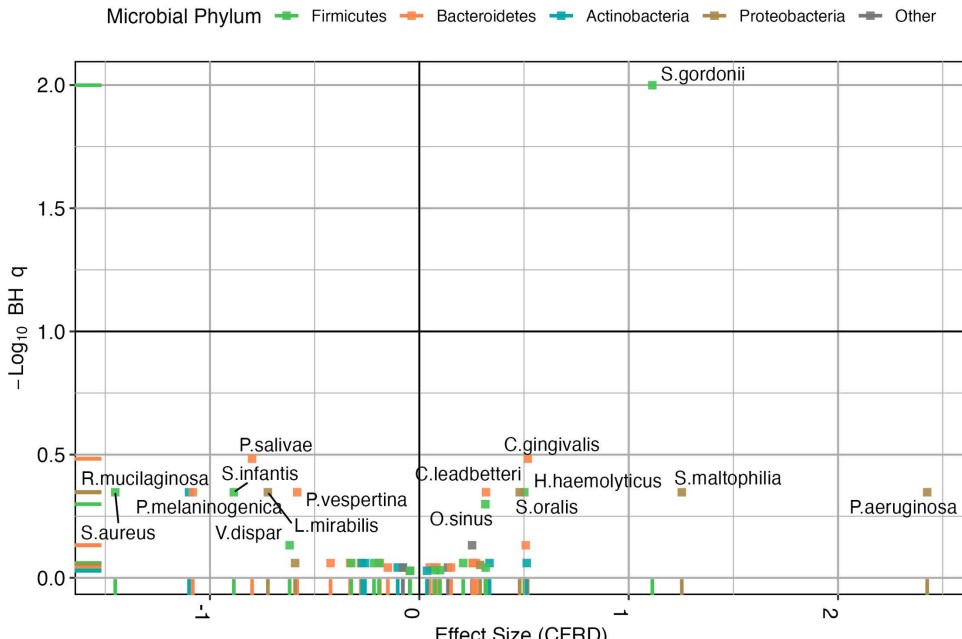

**Fig 2. Differentially abundant species in people with CFRD vs NGT.** The x-axis represents the effect size (regression coefficient) for CFRD status, while the y-axis shows the negative log10-transformed q-values. Each point represents a microbial taxon, colored by phylum.

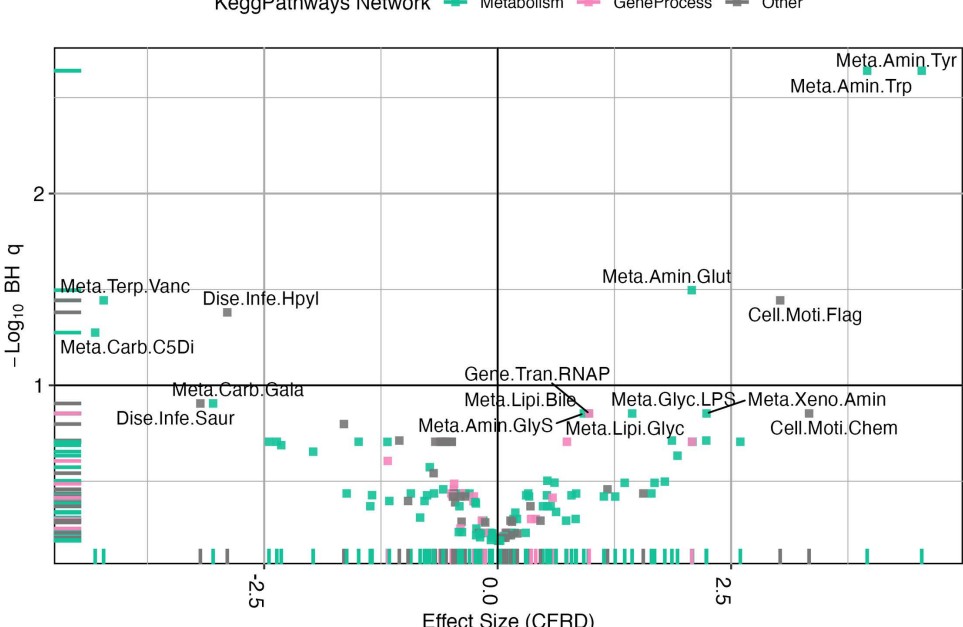

**Fig 3. Kegg Pathways difference in CFRD vs NGT.** The effect sizes of KEGG pathways in relation to cystic fibrosis-related diabetes (CFRD). The x-axis represents the effect size (regression coefficient) for CFRD status, while the y-axis shows the negative log10-transformed Benjamini-Hochberg (BH) corrected q-values. Each point corresponds to a KEGG pathway, colored by network classification (Metabolism, Gene Process, or Other). Dashed reference lines indicate significance thresholds, highlighting key pathways associated with CFRD.

abundance of various antimicrobial resistance genes (S3 Table). There was a nominal association between CFRD and the abundance of Phenicol resistance genes resulting in a log-fold increase of 2.03 (with a confidence interval of 0.90 to 3.16) in center-log ratio transformed relative abundance. Multiple other pathways of drug resistance are also nominally associated with CFRD status including lipopeptides, tetracycline resistance, multi drug efflux pumps and beta lactam resistance (Fig 4).

## Phage analysis

Phage analysis did not demonstrate a significant difference in viral species between pwCFRD and those with NGT.

## Viral and microbiome scores as predictors of CFRD

Microbial and viral signatures were generated using sPLS models with loadings plot showing the most influential species depicted in Fig 5. AUC for the microbiome score and virome score was 0.54 and 0.61 respectively.

To further validate the clinical relevance of these signatures, univariate and multivariate linear regression models incorporating both the virome and/or microbiome scores were used to predict lung function (FEV1 percent predicted). The CFRD virome signature score was associated with % predicted FEV1. In multivariate models adjusting for age, male sex, CFTR modulator use (Table 2, Model 3), a 10% increase in the virome signature score for CFRD was associated with a 5.7% decrease in % predicted FEV1. This association persisted even after adjusting for CFRD status (Model 5), suggesting that the airway virome signature has additional explanatory ability. In contrast, the CFRD microbiome signature score did not show a statistically significant association with lung function.

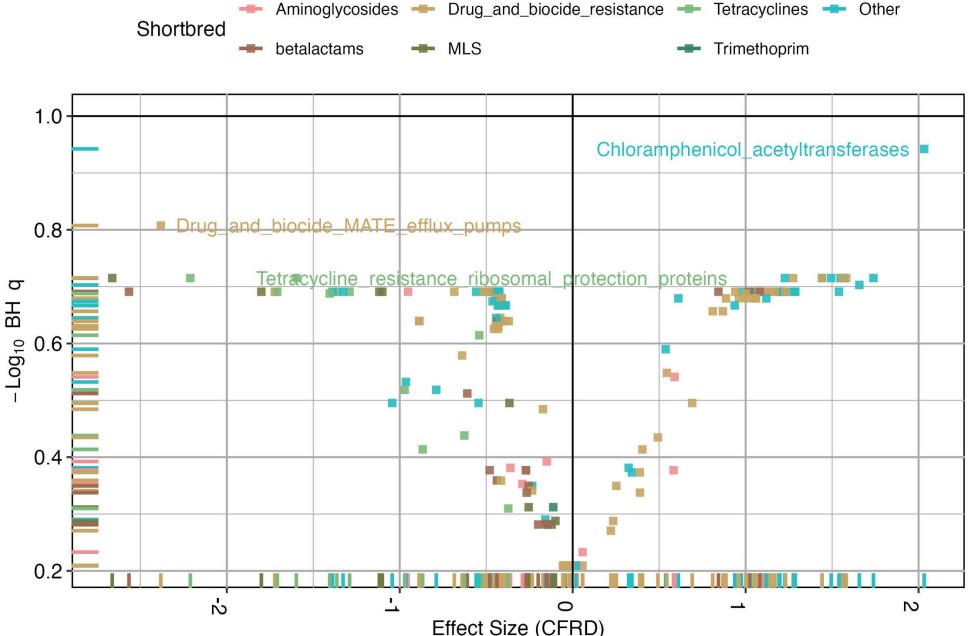

**Fig 4. Antibiotic resistance gene abundance differences in CFRD vs NGT.** The x-axis represents the effect size (regression coefficient) for CFRD status, while the y-axis shows the negative log10-transformed *q*-values. Each point corresponds to a specific resistance mechanism, colored by antibiotic class. Dashed reference lines indicate significance thresholds, highlighting key resistance mechanisms enriched in individuals with CFRD.

## Discussion

We demonstrated that airway microbial community structure and functional pathways are significantly different between pwCF with and without CFRD. The abundance of certain microorganisms, including important pathogens, may differ based on diabetes status. Additionally, there were observed differences in predicted microbial pathways in those with CFRD. These results provide preliminary data supporting the hypothesis that dysglycemia is associated with differences in the airway microbiome. Further, CFRD related virome signatures are independently associated with lung function. These hypotheses should be further investigated in future studies investigating the impact of CFRD on microbial diversity and antibiotic resistance patterns especially in longitudinal studies.

While hyperglycemia has been shown to predict culture-based pathogen detection in pwCF and in those with community acquired pneumonia [8], there are few studies evaluating the effect of hyperglycemia on the pulmonary microbiome. One study noted a difference in beta diversity in the bronchoalveolar lavage (BAL) of diabetic mice compared to non-diabetic littermate controls [27]. In SARS-CoV-2 infection in the non-CF population, people with diabetes had different beta diversities in nasopharyngeal microbiome compared to those without diabetes [28]. Additionally, taxonomic differences between the sputum microbiome in people with CFRD vs those without diabetes were identified in a small study using 16S rRNA sequencing. However, how the metagenomic microbiome may be impacted by glycemic status has not previously been studied in pwCF; therefore, this study represents an important advance in our understanding of how CFRD may lead to differences the lung microbiome, especially focusing on multi-kingdom differences at the species level and predicted microbial community functional differences which are only possible with metagenomic rather than amplicon sequencing.

We discovered differences in pathways and antibiotic gene profiles between the bacteria in sputum form pwCFRD vs NGT. The pathways found to be increased in the microbiome in CFRD could influence the interaction between the microbiome and host. The altered lung environment of individuals with CF and CFRD may lead to changes in nutrient availability.

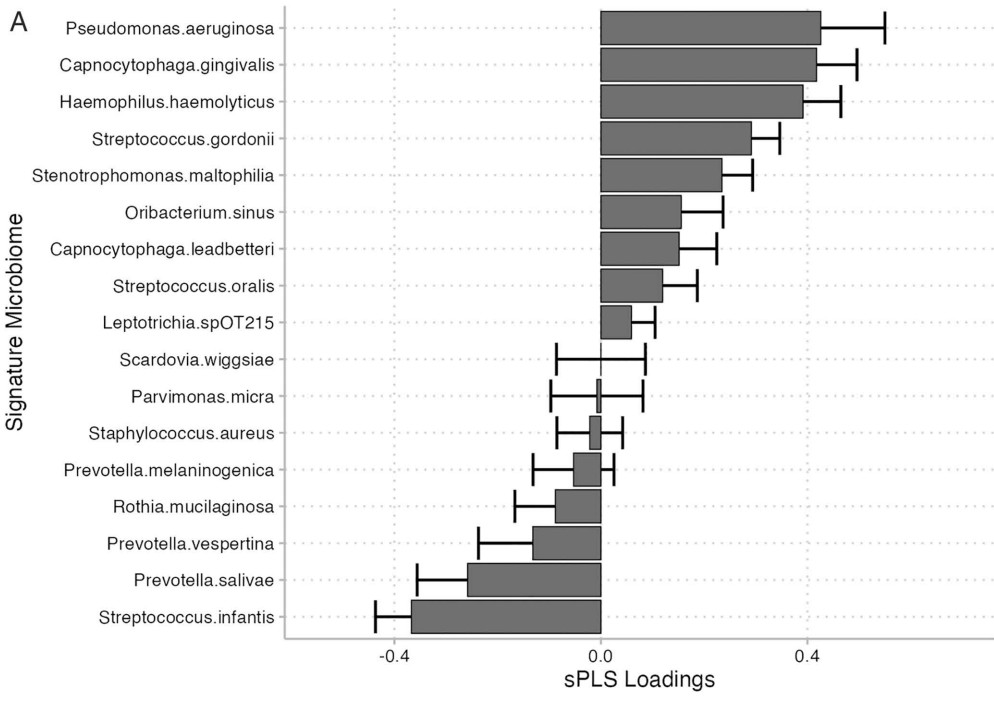

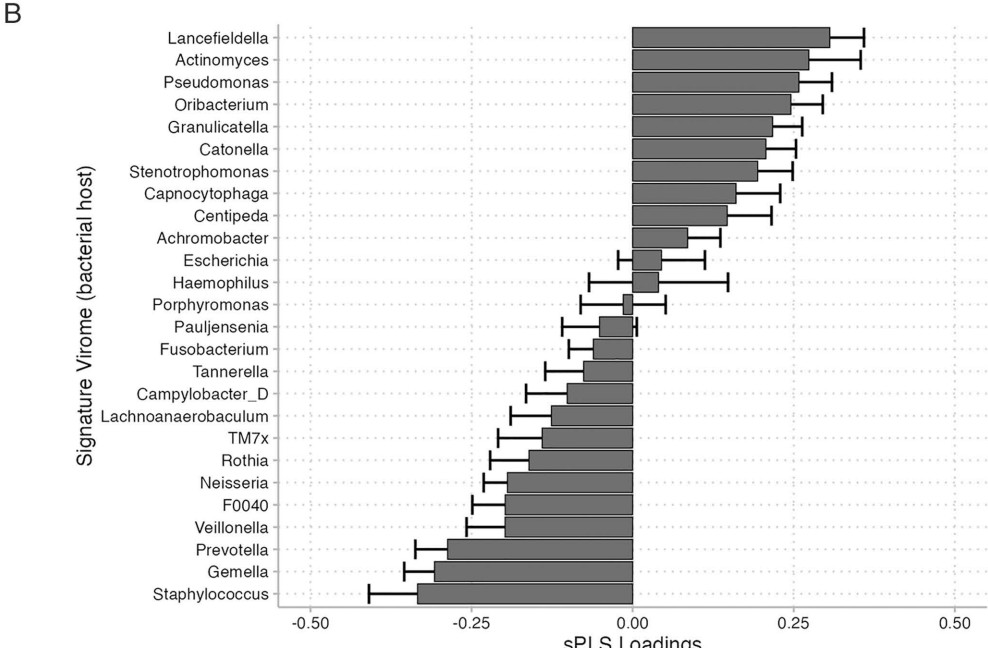

**Fig 5. Loading plot depicting influential species contributing to microbiome and virome signature for CFRD in sPLS model.** (A) Microbiome and (B) Virome profiles. Virome depicted here using predicted bacterial host to aid biological interpretation as they were largely tailed phages belonging to the *Caudoviricetes* class. The bar graphs illustrate the loadings for each microbial clade, with positive values indicating species associated with CFRD and negative values indicating species associated with NGT. Error bars represent the standard deviation of the loadings, providing insight into the variability within the models.

**Table 2. Association between microbiome and virome CFRD signatures and % predicted FEV1.**

| | Model 1 | Model 2 | Model 3 | Model 4 | Model 5 |
|---|---|---|---|---|---|
| Virome[a] | −6.1 (−9.2, −3.0)*** | – | −5.6 (−8.8, −2.4)** | – | −3.8 (−7.3, −0.3)* |
| Microbiome[a] | – | −2.9 (−6.3, 0.5) | – | −3.2 (−6.9, 0.5) | 0.6 (−2.9, 4.0) |
| Age, years | – | – | −0.2 (−0.7, 0.3) | −0.3 (−0.9, 0.4) | −0.2 (−0.6, 0.2) |
| Female sex | – | – | 3.9 (−9.7, 17.6) | 10.6 (−5.2, 26.5) | 13.2 (2.0, 24.5)* |
| CFTR Modulator | – | – | −1.0 (−16.3, 14.2) | 4.1 (−13.2, 21.4) | −2.6 (−15.7, 10.5) |
| CFRD status | – | – | −7.8 (−22.5, 6.9) | −7.1 (−24.7, 10.5) | 2.1 (−12.5, 16.6) |
| Antibiotics | – | – | −16.6 (−38.1, 4.8) | −25.3 (−52.9, 2.2) | −17.4 (−33.6, −1.3)* |

Univariate and multivariate linear regression models evaluating the association between CFRD-related microbiome and virome signatures and percent predicted FEV1. Model 1 includes only the virome score, while Model 2 includes only the microbiome score. Models 3 and 4 assess the virome and microbiome scores, respectively, with adjustment for age, sex, modulator use, CFRD status, and antibiotic use. Model 5 includes both the virome and microbiome scores in a single model, also adjusting for the same covariates. *$p < 0.05$; **$p < 0.01$; ***$p < 0.001$. [a]Virome and microbiome signature scores range from 0–100% with higher indicating more similar to CFRD and lower indicating more similar to NGT. Effect estimates provided for a 10% increase in score.

Our data show that bacterial populations in the CFRD sputum microbiome have increased pathways like amino acid metabolism and metabolism of cofactors and vitamins, potentially to adapt to altered nutrient sources. How these pathways interact and change virulence and influence of the microbial profiles will need further study. The discovery of antimicrobial resistance gene differences between pwCF with and without diabetes is a novel finding that could potentially impact clinical care and infectious disease management. However, it is import to note that although there is a correlation between the presence of antimicrobial resistance genes and clinical resistance to antibiotics, the accuracy of predicting resistance based on these genes varies depending on the specific pathogen [29]. This finding highlights the need for future studies to understand the etiology and clinical consequences of altered antibiotic resistance in CFRD. For example, the ARO 3005064 from the lipopeptide class is part of the CprRS system that confers adaptive resistance to colistin antibiotics [30], an antibiotic commonly used in pwCF. This finding is an example of the emerging role of genomics in tailoring medical interventions and underscores the promise of precision medicine in combating infectious diseases in vulnerable populations.

The CFRD-associated virome score emerged as associated with percent predicted FEV1, whereas the microbiome score was not. The lack of association between the microbiome score and lung function suggests that phage activity is not simply a reflection of bacterial composition but may represent an independent factor influencing disease severity. Previous studies that have looked at the virome profiles in asthma have shown lower lung function in people with certain virome characteristics [31]. Phages play a critical role in shaping bacterial communities through predation, horizontal gene transfer, and lysogenic conversion, which can modulate bacterial virulence, antibiotic resistance, and immune interactions. Additionally, phages may directly influence host immune responses, contributing to airway inflammation and epithelial damage. These findings highlight the need to further investigate the role of phages in CF lung disease and consider phage-targeted interventions as a potential therapeutic approach. Future studies should explore how phage-host interactions contribute to pulmonary decline and whether modulating the airway phageome could provide new strategies for preserving lung function in CF and CFRD.

Limitations of this study warrant consideration. The modest sample size limits the power for differential abundance analysis and to examine effect modifiers such as use of glucocorticoids. This was a cross-sectional study; while we have prospective lung function data on participants, the period after sputum collection and initial lung function testing was confounded by both the Covid19 pandemic as well as the initiation of CFTR modulators for a substantial minority of participants. In addition, more comprehensive glycemic characterization with prospective oral glucose tolerance testing

and continuous glucose monitoring will be important in future studies. The virome and microbiome models have modest prediction value for CFRD likely due to the individualized nature of each person's microbiome as well the small sample size. However, the depth of our sequencing, our use of metagenomic sequencing to highlight functional differences, and the deep phenotyping represent important strengths of our analyses.

In conclusion, CFRD is associated with global changes to the airway microbiome in pwCF. Virome signatures of CFRD are associated with worse lung function. Not only is hyperglycemia associated with worse outcomes in CF, but the same holds true for other pulmonary diseases such as pneumonia and COVID-19 [8]. Future studies with rigorous characterization of glycemic status, longitudinal follow-up, and multi-kingdom characterization of microbial ecology in CFRD would be important in determining whether glycemic control may serve as a viable therapy for management of lung disease.

## Supporting information

**S1 Table. Differential abundance of top significant taxa comparing CFRD vs NGT.**
(DOCX)

**S2 Table. Differential impact of CFRD on predicted functional pathways.**
(DOCX)

**S3 Table. Antimicrobial resistance genes abundance comparing CFRD vs NGT.**
(DOCX)

**S1 File. Supplementary methods.**
(DOCX)

## Author contributions

**Conceptualization:** Laura N. Brenner, Peggy S. Lai.

**Data curation:** Laura N. Brenner, Ching-Ying Huang, Leonard Sicilian.

**Formal analysis:** Laura N. Brenner, Ching-Ying Huang, Minsik Kim, Peggy S. Lai.

**Funding acquisition:** Laura N. Brenner, Melissa S. Putman.

**Investigation:** Laura N. Brenner, Minsik Kim, Christopher J. Richards, Lindsay Bringhurst, Isabel Neuringer.

**Methodology:** Minsik Kim, Leonard Sicilian.

**Project administration:** Lindsay Bringhurst, Isabel Neuringer, Peggy S. Lai.

**Supervision:** Melissa S. Putman, Peggy S. Lai.

**Visualization:** Ching-Ying Huang.

**Writing – original draft:** Laura N. Brenner.

**Writing – review & editing:** Ching-Ying Huang, Christopher J. Richards, Melissa S. Putman, Peggy S. Lai.

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
