## [Decision Letter · Decision Letter 0]

5 Jun 2025

PONE-D-25-12412Dysglycemia and the airway microbiome in cystic fibrosis.PLOS ONE

Dear Dr. BRENNER,

Thank you for submitting your manuscript to PLOS ONE. After careful consideration, we feel that it has merit but does not fully meet PLOS ONE’s publication criteria as it currently stands. Therefore, we invite you to submit a revised version of the manuscript that addresses the points raised during the review process.

We look forward to receiving your revised manuscript.

Kind regards,

Vinod Kumar Yata, PhD

Academic Editor

PLOS ONE

**Journal Requirements:**

1. When submitting your revision, we need you to address these additional requirements. Please ensure that your manuscript meets PLOS ONE's style requirements, including those for file naming. The PLOS ONE style templates can be found at https://journals.plos.org/plosone/s/file?id=wjVg/PLOSOne_formatting_sample_main_body.pdf and https://journals.plos.org/plosone/s/file?id=ba62/PLOSOne_formatting_sample_title_authors_affiliations.pdf 2. Thank you for stating in your Funding Statement: L.N.B is supported by the NIDDK K23 DK125839 and CFF K Boost award. Please provide an amended statement that declares *all* the funding or sources of support (whether external or internal to your organization) received during this study, as detailed online in our guide for authors at http://journals.plos.org/plosone/s/submit-now.  Please also include the statement “There was no additional external funding received for this study.” in your updated Funding Statement. Please include your amended Funding Statement within your cover letter. We will change the online submission form on your behalf. 3. Please note that your Data Availability Statement is currently missing the direct link to access each database. If your manuscript is accepted for publication, you will be asked to provide these details on a very short timeline. We therefore suggest that you provide this information now, though we will not hold up the peer review process if you are unable. 4. Please amend either the abstract on the online submission form (via Edit Submission) or the abstract in the manuscript so that they are identical. 5. We notice that your supplementary tables are included in the manuscript file. Please remove them and upload them with the file type 'Supporting Information'. Please ensure that each Supporting Information file has a legend listed in the manuscript after the references list.

Reviewers' comments:

Reviewer's Responses to Questions

**Comments to the Author**

1. Is the manuscript technically sound, and do the data support the conclusions?

Reviewer #1: Yes

2. Has the statistical analysis been performed appropriately and rigorously? 

Reviewer #1: Yes

3. Have the authors made all data underlying the findings in their manuscript fully available?

Reviewer #1: Yes

4. Is the manuscript presented in an intelligible fashion and written in standard English?

Reviewer #1: Yes

5. Review Comments to the Author

**Reviewer #1: ** The authors attempt to associate cystic fibrosis related diabetes condition to changes in the airway microbiome using metagenomics which as the authors have evidenced in this study and analysis, an important factor in understanding this lung disease. The study design is sound with recruitment, sample and clinical data collection, group stratification based on glucose tolerance, etc. The statistical analysis performed is also very appropriate with the metrics used widely in ecology studies (diversity indices, species abundance, species richness, etc.). The preliminary conclusions seem to be supported by the data and overall this paper seems complete with a note in the discussion on future directions with more rigorous strategies on glycemic characterization, longitudinal follow-up, etc.

Only a few minor comments follow:

1. I understand the logic behind use of inverse Simpson index to assess species diversity and Bray-Curtis distance for dissimilarity in species composition was very intuitive. It would be great if the data were also tested on another diversity index since in my experience the sensitivity of true diversity to rare vs abundant species varies in different indices. Although not crucial, but this would be good to verify the conclusions of the study.

2. The value of inverse Simpson indicating a lower alpha diversity associated with CFRD status in results section (-2.30) is different than the one given in the abstract (-1.98) although confidence intervals and p values agree.

3. In Figure 4 showing Antibiotic resistance gene abundance differences, the labels overlap a lot so hard to read. Also the figure uses BH corrected q values on y axis but the figure description mentions using raw p values instead. Also a small justification of why figure 2 used raw p values whereas in 3 and 4, BH q values were used.

4. In Figure 5a and 5b, is there a specific reason that the bars are different colors? They don't match other figures which have their own legends which makes complete sense if colors are different, but the lack of a legend in 5a and 5b suggests that the bars really don't need to be colored.

5. Table 2 which mentions association between microbiome and virome CFRD signatures and percent predicted FEV1 seems like the main table supporting the conclusion of the paper that virome and not microbiome being associated to lung function. This table was hard for me to really understand what the different regression models were in terms of which combinations of confounders were tested so a brief explanation on how models 1 through 5 are different would be nice.

Apart from these, minor typos in the discussion sections and where Trimmomatic is cited in the methods section. Overall, a fun and exciting paper to review.

6. PLOS authors have the option to publish the peer review history of their article (what does this mean? ). If published, this will include your full peer review and any attached files.

**Do you want your identity to be public for this peer review?** For information about this choice, including consent withdrawal, please see our Privacy Policy .

Reviewer #1: **Yes: ** Ruchit Panchal

---

## [Author Response · Author response to Decision Letter 1]

27 Jun 2025

We greatly appreciate the thoughtful review of our manuscript. We have made numerous revisions in response to these comments, which we believe have improved the quality of the manuscript. We describe these in detail below.

Reviewer #1: The authors attempt to associate cystic fibrosis related diabetes condition to changes in the airway microbiome using metagenomics which as the authors have evidenced in this study and analysis, an important factor in understanding this lung disease. The study design is sound with recruitment, sample and clinical data collection, group stratification based on glucose tolerance, etc. The statistical analysis performed is also very appropriate with the metrics used widely in ecology studies (diversity indices, species abundance, species richness, etc.). The preliminary conclusions seem to be supported by the data and overall this paper seems complete with a note in the discussion on future directions with more rigorous strategies on glycemic characterization, longitudinal follow-up, etc.

Response: Thank you for your generous comments and suggestions.

Only a few minor comments follow:

1. I understand the logic behind use of inverse Simpson index to assess species diversity and Bray-Curtis distance for dissimilarity in species composition was very intuitive. It would be great if the data were also tested on another diversity index since in my experience the sensitivity of true diversity to rare vs abundant species varies in different indices. Although not crucial, but this would be good to verify the conclusions of the study.

Response: We appreciate this thoughtful suggestion regarding the use of additional diversity indices. We have done so and the results remain consistent. We revised the results as below:

In our linear model assessing the relationship between glycemic status and alpha diversity, CFRD status was associated with lower alpha diversity, as reflected by a decrease in the Inverse Simpson index (-1.98 [-3.80, -0.16], p = 0.033) and a nonsignificant reduction in species richness (-3.13 [-7.61, 1.34], p = 0.17). In the multivariate model, the effect size of the association between CFRD status and the Inverse Simpson index remained similar, with a beta estimate of -2.00 [-4.02, 0.02] (p = 0.052). IGT status was not significantly associated with alpha diversity. To test the hypothesis that glycemia affected overall microbial community structure, we assessed differences in beta diversity. CFRD status was associated with microbial community structure as measured by the Bray Curtis (PERMANOVA p=0.011, R2= 17%) (Figure 1) and Horn-Morisita (PERMANOVA p= 0.0067, R2 = 19%) dissimilarity in the multivariate model.

2. The value of inverse Simpson indicating a lower alpha diversity associated with CFRD status in results section (-2.30) is different than the one given in the abstract (-1.98) although confidence intervals and p values agree.

Response: We apologize for the inconsistency. The results section reported an incorrect effect size and was updated to the correct one.

3. In Figure 4 showing Antibiotic resistance gene abundance differences, the labels overlap a lot so hard to read. Also the figure uses BH corrected q values on y axis but the figure description mentions using raw p values instead. Also a small justification of why figure 2 used raw p values whereas in 3 and 4, BH q values were used.

Response: We have updated figure 4 to have less overlap and the updated the legend to use q-values. We have updated all the figures to use q values for consistency.

4. In Figure 5a and 5b, is there a specific reason that the bars are different colors? They don't match other figures which have their own legends which makes complete sense if colors are different, but the lack of a legend in 5a and 5b suggests that the bars really don't need to be colored.

Response: We appreciate this helpful suggestion. We agree that the color variation in Figures 5a and 5b was unnecessary and potentially confusing. We have now updated both figures to use a single, consistent color for all bars. Additionally, we have included updated loading plots that incorporate covariates for clarity and completeness. These changes have improved the consistency and interpretability of the figure.

5. Table 2 which mentions association between microbiome and virome CFRD signatures and percent predicted FEV1 seems like the main table supporting the conclusion of the paper that virome and not microbiome being associated to lung function. This table was hard for me to really understand what the different regression models were in terms of which combinations of confounders were tested so a brief explanation on how models 1 through 5 are different would be nice.

Response: Thank you for point this out. We have updated the table legend to better describe each models as below:

"Univariate and multivariate linear regression models evaluating the association between CFRD-related microbiome and virome signatures and percent predicted FEV1. Model 1 includes only the virome score, while Model 2 includes only the microbiome score. Models 3 and 4 assess the virome and microbiome scores, respectively, with adjustment for age, sex, modulator use, CFRD status, and antibiotic use. Model 5 includes both the virome and microbiome scores in a single model, also adjusting for the same covariates."

6. Apart from these, minor typos in the discussion sections and where Trimmomatic is cited in the methods section. Overall, a fun and exciting paper to review.

Response: Thank you! We have fixed the typos.

---

## [Editor Report · Decision Letter 1]

22 Aug 2025

Dysglycemia and the airway microbiome in cystic fibrosis.

PONE-D-25-12412R1

Dear Dr. BRENNER,

We’re pleased to inform you that your manuscript has been judged scientifically suitable for publication and will be formally accepted for publication once it meets all outstanding technical requirements.

Kind regards,

Vinod Kumar Yata, PhD

Academic Editor

PLOS ONE
---

## [Editor Report · Acceptance letter]

PONE-D-25-12412R1

PLOS ONE

Dear Dr. Brenner,

I'm pleased to inform you that your manuscript has been deemed suitable for publication in PLOS ONE. Congratulations! Your manuscript is now being handed over to our production team.

Kind regards,

on behalf of

Dr. Vinod Kumar Yata

Academic Editor

PLOS ONE